# The COVID-19 pandemic effect on the prehospital Madrid stroke code metrics and diagnostic accuracy

Nicolás Riera-López[1]*, Andrea Gaetano-Gil[2], José Martínez-Gómez[3], Nuria Rodríguez-Rodil[3], Borja M. Fernández-Félix[2,4], Jorge Rodríguez-Pardo[5], Carmen Cuadrado-Hernández[1], Emmanuel Pelayo Martínez-González[1], Alicia Villar-Arias[6], Fátima Gutiérrez-Sánchez[6], Pablo Busca-Ostolaza[6], Eduardo Montero-Ruiz[7], Exuperio Díez-Tejedor[5], Javier Zamora[2,4,8‡], Blanca Fuentes-Gimeno[5‡], on behalf of the Madrid Stroke Network[¶]

1 Stroke Commission, Emergency Medical Service of Madrid (SUMMA 112), Madrid, Spain, 2 Clinical Biostatistics Unit, Ramón y Cajal University Hospital, IRYCIS, Madrid, Spain, 3 IT Department, Emergency Medical Service of Madrid (SUMMA 112), Madrid, Spain, 4 CIBER of Epidemiology and Public Health (CIBERESP), Madrid, Spain, 5 Department of Neurology and Stroke Centre, IdiPAZ Health Research Institute (La Paz University Hospital, Autonomous University of Madrid), Madrid, Spain, 6 Management Department, Emergency Medical Service of Madrid (SUMMA 112), Madrid, Spain, 7 Internal Medicine Department, Ramón y Cajal University Hospital, Madrid, Spain, 8 WHO Collaborating Centre for Global Women's Health, Institute of Metabolism and Systems Research, University of Birmingham, Birmingham, United Kingdom

‡ JZ and BFG are co-last authors on this work.
¶ Membership of the Madrid Stroke Network is provided in Acknowledgments and in the S1 File.
* nicolas.riera@salud.madrid.org

**Data Availability Statement:** Data belongs to our Regional Health Authority, and each of the participant hospitals. Spanish law requires

## Abstract

### Background

Only very few studies have investigated the effect of the COVID-19 pandemic on the pre-hospital stroke code protocol. During the first wave, Spain was one of the most affected countries by the SARS-CoV-2 coronavirus disease pandemic. This health catastrophe over-shadowed other pathologies, such as acute stroke, the leading cause of death among women and the leading cause of disability among adults. Any interference in the stroke code protocol can delay the administration of reperfusion treatment for acute ischemic strokes, leading to a worse patient prognosis. We aimed to compare the performance of the stroke code during the first wave of the pandemic with the same period of the previous year.

### Methods

This was a multicentre interrupted time-series observational study of the cohort of stroke codes of SUMMA 112 and of the ten hospitals with a stroke unit in the Community of Madrid. We established two groups according to the date on which they were attended: the first during the dates with the highest daily cumulative incidence of the first wave of the COVID-19 (from February 27 to June 15, 2020), and the second, the same period of the previous year (from February 27 to June 15, 2019). To assess the performance of the stroke code, we compared each of the pre-hospital emergency service time periods, the diagnostic accuracy

permission to be obtained from each of the data protection delegates (contact information: Ricardo Iglesias, email: protecciondedatos. sanidad@madrid.org) to share data with third parties.

**Funding:** NRL, Order HAC/667/2020, Foundation for Biosanitary Research and Innovation in Primary Care (FIIBAP), https://www.fiibap.org/. The funders had no role in study design, data collection and analysis, decision to publish, or preparation of the manuscript.

**Competing interests:** I have read the journal's policy and the authors of this manuscript have the following competing interests: Nicolás Riera-López and Jorge Rodríguez-Pardo de Donlebún have received payments for training courses from the Angels Initiative (Boheringer Ingelheim). The other authors do not report any conflicts of interest. This does not alter our adherence to PLOS ONE policies on sharing data and materials.

**Abbreviations:** CM, Community of Madrid; COVID-19, Disease caused by the new coronavirus 2019; SARS-CoV-2, Severe acute respiratory syndrome coronavirus 2; AS, Acute stroke; SUMMA-112, Madrid Emergency Medical Service 112; SC, Stroke code; EMS, Emergency Medical Service; IVT, Intravenous thrombolysis; MBDS, Minimum Basic Dataset; M-Direct, Madrid-Direct scale; MT, Mechanical thrombectomy; SU, Stroke Unit; HIN, Health Identification Number.

(proportion of stroke codes with a final diagnosis of acute stroke out of the total), the proportion of patients treated with reperfusion therapies, and the in-hospital mortality.

## Results

SUMMA 112 activated the stroke code in 966 patients (514 in the pre-pandemic group and 452 pandemic). The call management time increased by 9% (95% CI: -0.11; 0.91; p value = 0.02), and the time on scene increased by 12% (95% CI: 2.49; 5.93; p value = <0.01). Diagnostic accuracy, and the proportion of patients treated with reperfusion therapies remained stable. In-hospital mortality decreased by 4% (p = 0.05).

## Conclusions

During the first wave, a prolongation of the time "on the scene" of the management of the 112 calls, and of the hospital admission was observed. Prehospital diagnostic accuracy and the proportion of patients treated at the hospital level with intravenous thrombolysis or mechanical thrombectomy were not altered with respect to the previous year, showing the resilience of the stroke network and the emergency medical service.

---

## Introduction

Spain was one of the most affected countries by the global pandemic caused by the coronavirus SARS-CoV-2 (COVID-19). By June 15, 2020, 70,554 cases and 9,157 deaths were reported in the Community of Madrid (CM) [1, 2]; however, both figures are likely to be underestimations, given the low number of SARS-CoV-2 detection tests performed in the initial stages of the pandemic [3, 4]. This health catastrophe has overshadowed other pathologies, producing considerable interference in health systems.

Acute stroke (AS) is the second leading cause of global mortality and the leading cause of disability in adults [5]. It is the time-dependent pathology that is most attended to in the Emergency Medical Service of Madrid (SUMMA 112) [6]. The treatments developed over the last three decades have improved mortality and the degree of dependence suffered by patients with acute ischaemic stroke, but the outcomes depend on the time that elapses from the onset of symptoms to the administration of reperfusion treatment. It has been estimated that each minute saved in the start of those treatments adds approximately one week of life without sequelae [7].

The detection of calls reporting suspicious symptoms of AS, the dispatch of an ambulance with the highest priority, the in-situ assessment of the patient, the selection of the appropriate hospital (the nearest one that has available the treatment that the patient may require) alerting the neurologist on duty, and the rapid transfer of the patient constitute the core aspects of the "stroke code" (SC) protocol. Its implementation has led to a significant reduction in time to treatments, with consequent impacts on the patient's outcomes [8–10]. Therefore, international guidelines recommend accessing the health system through the emergency medical service (EMS) by calling 112 [11, 12]. Any interference with the functionality of this process can impact stroke care in general, including the delay the administration of reperfusion treatment in ischemic stroke patients, and therefore worsen the patient's outcomes.

During the first wave of the pandemic, several groups warned about a drastic reduction in cases of AS admitted to the hospitals, the saturation of call centres, delays in ambulance service

and even saturation of hospital emergency departments [13–16]. However, there is not enough information on the specific aspects that have been most affected in the prehospital and hospital phases of urgent care for AS. According to the recommendations of researchers and international organizations, a rigorous examination of what has occurred during the COVID-19 pandemic is necessary to issue recommendations that improve the stability of the protocol in times of crisis [17, 18].

The main aim of the study was to compare the periods of time spent in each of the phases of the prehospital SC in the CM before the onset of the pandemic and during the period with the highest incidence of SARS-CoV-2 infection in the first wave. The secondary aim was to compare the other fundamental aspects of the functioning of the SC between the same periods (among which is the diagnostic accuracy of the EMS, the proportion of patients treated with reperfusion therapies, and in-hospital mortality).

## Materials and methods

An interrupted time series multicentre observational study was conducted that included the SC cohort of SUMMA 112 and the discharge summary data of the ten hospitals with a Stroke Unit of the Madrid Health Region. The protocol was approved by the Drug Ethics Committee of the CM (Act 12/2020), which authorized the exemption of informed consent. The anonymized and aggregated data is available upon request.

### Setting and population

SUMMA 112 is the main EMS in the CM, with a population of 6.5 million inhabitants in 2020. Annually, the SC protocol is activated in approximately 2,000 patients who meet the following criteria: symptoms suggestive of stroke, lasting less than 24 hours or unknown onset, and absence of significant previous functional dependence.

The call to the 112-emergency number is handled in two steps. The first person, the telephone operator, collects the location of the incident and the main symptom in a logic tree. The call is then passed to a physician who conducts a structured interview and provides initial instructions to the patient. If the stroke code activation criteria are met, an advanced ambulance (with an emergency physician) is dispatched to assess the patient in situ. Then, the destination hospital is selected using the Madrid-Direct scale (M-Direct), which has shown a high sensitivity and specificity to identify at prehospital levels patients with large vessel occlusion who could benefit from treatment with mechanical thrombectomy (MT) [19] (S2 File). And finally, the neurologist on duty is alerted, and the ambulance makes the urgent transfer to that hospital. The pre-hospital SC protocol has not changed during the period of the study.

The Stroke Network of the CM is composed of hospitals classified by levels according to their capacity. Ten of them have a stroke Unit (SU) and can provide intravenous thrombolysis (IVT). Seven are MT-ready on a weekly rotation basis, ensuring that three hospitals provide full-time coverage every day [20, 21]. Patients are evaluated by the on-duty neurologist, who administers urgent treatment and admits them to the stroke unit after performing the relevant imaging tests. Finally, at hospital discharge, a report is filled in with all the diagnoses and treatments, which are recorded by the Coding Service of each hospital in the Minimum Basic Data Set Dataset (MBDS).

The first case of COVID-19 in the CM was reported on February 25, 2020. During the first wave of the pandemic, the median cumulative daily incidence was 98 cases of COVID-19 per 100,000 inhabitants, detected by antigen testing [1]. Because the objective is to analyse what happened at the times of highest incidence, the days on which the reported ones were above

the median of the daily cumulative incidence during the first wave were selected for the pandemic group.

**Inclusion criteria.** Patients meeting SC criteria. The recruitment period was split into the following dates: pandemic group (February 27 to June 15, 2020, 79 days) and pre-pandemic group (the same period of the previous year, February 27 to June 15, 2019, 78 days).

**Exclusion criteria.** Patients without Health Identification Number (HIN), or without a record in the MBDS.

## Data collection

All the prehospital variables were collected from the clinical report on the tablet pc, that feeds the prospective SC database of SUMMA 112. The hospital variables were collected from the MBDS. It is an administrative registry that records the diagnoses and treatments reflected in the hospital discharge report [22]. Both databases were linked using a unidirectionally ciphered field of the HIN to ensure that the information corresponded to the same patient [23].

## Variables

For the main objective, time elapsed in each of the steps of prehospital care were analysed (Fig 1).

Times a and b are automatically recorded by the system. Times c, d, e and f are manually recorded by pressing a button on the ambulance's tablet PC. Finally, time g is manually recorded by the MDBS Coding Service with the date and time of the hospital discharge report.

For the secondary objective, data on clinical, process and outcome variables of the prehospital and hospital phases of SC were collected, according to international recommendations for quality control [11, 12, 24].

In the prehospital phase, the following data were collected: age, sex, vital signs (blood pressure, heart rate, respiratory rate, oxygen saturation, glycaemia, temperature, and electrocardiographic rhythm), suspicion of large vessel occlusion (measured with the M-Direct scale, see S2 File), if it was cannulated an intravenous line, place of patient admission and Glasgow Coma Scale.

In the hospital phase, the diagnostic accuracy was calculated as the proportion of SC with a final diagnosis of AS (as defined in S1 Table) out of the total number of records, without distinguishing between ischemic and haemorrhagic stroke. The overall severity status of the patients was also analysed using three parameters: the Charlson comorbidity index with updated weights (which evaluates life expectancy at 10 years, classifying the patient's comorbidity as absent, low or high) [25, 26]; the degree of severity and risk of mortality (which

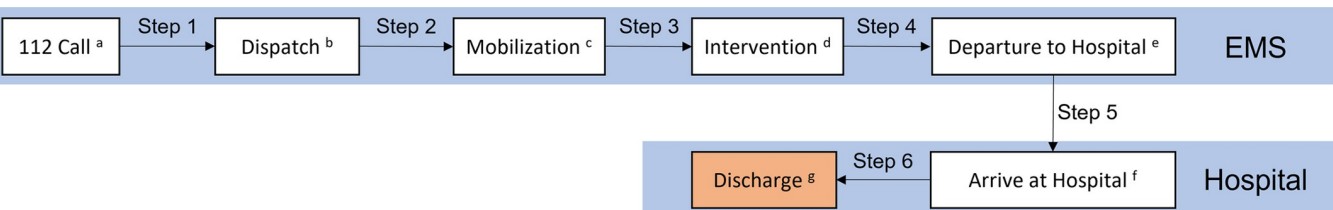

**Fig 1. Flowchart of the time periods analysed.** [a] Call is picked up by the 112 operator. [b] A message with the mission data is sent to the ambulance. [c] Ambulance starts moving towards the mission address. [d] Ambulance stops upon arrival at the mission address. [e] Ambulance with the patient begins to move towards the hospital. [f] Ambulance with the patient stops upon arrival at the hospital. [g] Patient is discharged from the hospital (to home or another health centre).

predict in-hospital mortality with four degrees of risk, namely, minor, moderate, major or extreme) [27], the reperfusion treatment applied, and in-hospital mortality.

## Statistical analysis

Categorical variables are presented as absolute and relative frequencies, quantitative variables with a normal distribution are presented as means and standard deviations, and variables that do not follow a normal distribution are presented as medians and interquartile ranges. Differences between the two analysis groups were evaluated using Student's t-tests or Mann–Whitney U tests and chi-square tests, as appropriate.

To evaluate the magnitude of the pandemic effect we calculated the difference between medians of the pre-pandemic and pandemic groups. The confidence intervals for these differences in medians were estimated using non-parametric bootstrap techniques with 800 resamples [28]. The time distributions in both periods are shown graphically in the form of boxplots. To evaluate the direct effect of the pandemic on intervention times, as well as to analyse eventual temporal trends, an interrupted time series analysis was carried out using linear regression models. The slopes of the pre-pandemic and pandemic periods and the effect of the pandemic on February 27, 2020, are reported with their respective 95% confidence intervals. Only differences with $p < 0.05$ were considered statistically significant. Statistical software Stata v16 (StataCorp LLC, College Station, Texas) was used for the analyses.

## Results

SUMMA 112 activated the SC in 966 patients (514 in the pre-pandemic group and 452 pandemic). For 88 (17%) patients in the pre-pandemic period and 83 (18%) in the pandemic period, the corresponding hospital MBDS record was not identified. Fig 2 shows the flowchart of the patients. No repeated cases were found.

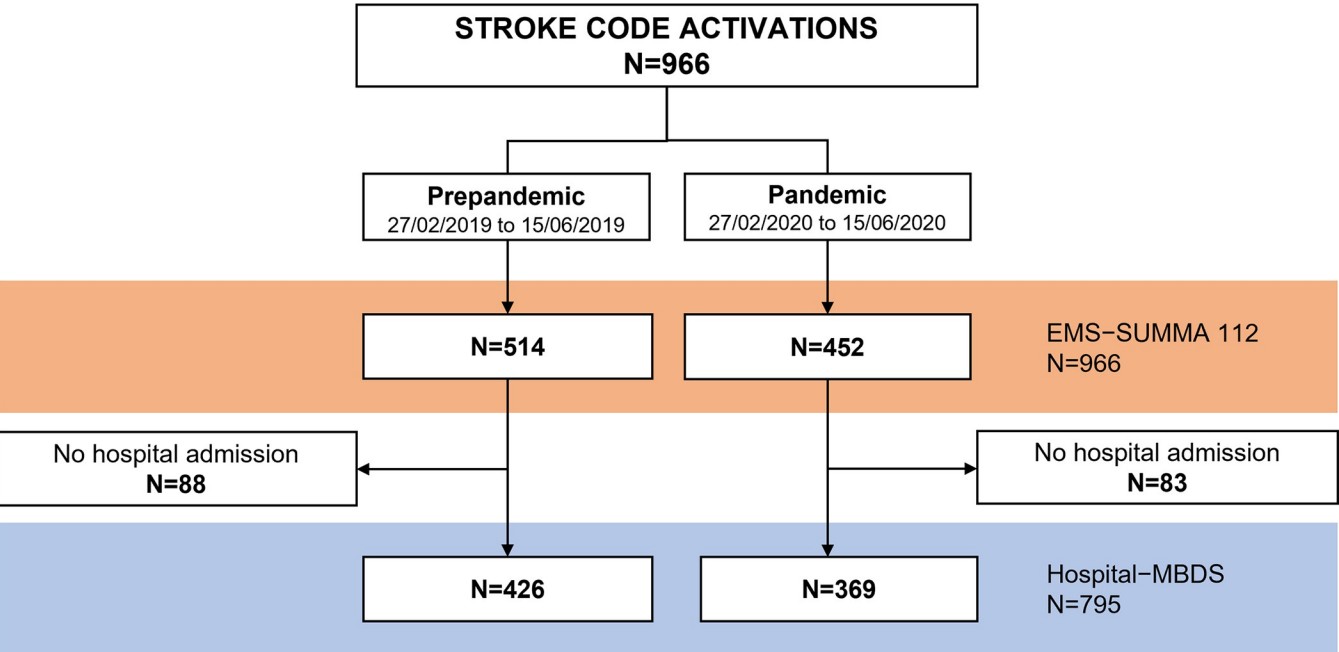

**Fig 2. Patient flowchart.** SUMMA 112: EMS of Madrid. MBDS: Minimum Basic Dataset.

Table 1 present the baseline characteristics of the patients treated by EMS. During the pandemic, 6.4% fewer SCs were activated compared to the pre-pandemic period. Patients were younger (70 vs. 72 years, difference 2.2; 95% CI: 0.29; 4.0; p value = 0.02) and comprised a

**Table 1. Baseline characteristics.**

| Prehospital variables | $N_0|N_1$ | Pre-pandemic | Pandemic | P value |
|---|---|---|---|---|
| Sex: Men n (%) | 514|452 | 243 (47.3) | 243 (53.8) | 0.044 |
| Age (years) | 514|452 | 72.4 (14.4) | 70.2 (15.0) | 0.012 |
| Over 80 years n (%) | | 182 (35.4) | 128 (28.3) | 0.020 |
| Systolic blood pressure (mmHg) | 492|444 | 155.9 (28.7) | 155.3 (30.5) | 0.574 |
| Diastolic blood pressure (mmHg) | 492|446 | 87.2 (18.9) | 87.7 (20.1) | 0.840 |
| Heart rate (bpm) | 491|438 | 84.6 (23.7) | 83.9 (22.2) | 0.794 |
| Respiratory rate (rpm) | 237|310 | 15.8 (4.8) | 15.5 (4.1) | 0.520 |
| O2 saturation (%) | 452|418 | 94.8 (5.0) | 94.8 (4.1) | 0.971 |
| Blood glucose level (mg/dl) | 482|426 | 134.2 (47.1) | 134.4 (47.1) | 0.814 |
| Temperature (˚C) | 425|392 | 35.95 (0.6) | 35.97 (0.7) | 0.513 |
| Electrocardiogram n(%) | 470|375 | 415 (88.3) | 293 (78.1) | <0.001 |
| Madrid-Direct scale n (%) | 282|322 | | | |
| Positives (M-Direct scale S≥2) | | 76 (14.8) | 104 (23) | 0.150 |
| Rhythm n (%) | 407|339 | | | |
| AV block | | 5 (1.2) | 13 (3.8) | 0.130 |
| Atrial fibrillation | | 81 (19.9) | 59 (17.4) | |
| Pacemaker Rhythm | | 15 (3.7) | 9 (2.7) | |
| Sinus | | 304 (74.7) | 252 (74.3) | |
| Other | | 2 (1.1) | 6 (1.8) | |
| Intravenous line n (%) | 471|376 | 422 (89.6) | 317 (84.3) | 0.022 |
| Place of patient admission n (%) | 228|259 | | | |
| Critical emergency room | | 174 (76.3) | 223 (86.1) | <0.001 |
| Other | | 54 (23.7) | 36 (13.9) | |
| Glasgow Coma Scale | 464|420 | 13.2 (2.7) | 13.3 (2.6) | 0.961 |
| **Hospital variables** | | Pre-pandemic | Pandemic | P value |
| Reperfusion treatments n (%) | | | | |
| Mechanical thrombectomy | 426|369 | 83 (19.9) | 69 (18.9) | 0.741 |
| Intravenous thrombolysis | 426|369 | 61 (16.7) | 70 (16.8) | 0.999 |
| Number of concomitant diseases n (%) | 426|369 | 11.8 (4.8) | 12.2 (4.5) | 0.333 |
| Charlson comorbidity index (xx) | 426|369 | 1.6 (2.6) | 1.8 (4.0) | 0.873 |
| Diagnosis related groups weight | 426|369 | 1.4 (1.6) | 1.2 (1.1) | 0.660 |
| Predicted risk of mortality n (%) | 426|369 | | | |
| Minor | | 91 (21.4) | 83 (22.5) | 0.050 |
| Moderate | | 207 (48.6) | 168 (45.5) | |
| Major | | 70 (16.4) | 84 (22.8) | |
| Extreme | | 58 (13.6) | 34 (9.2) | |
| Degree of severity n (%) | 426|369 | | | |
| Minor | | 66 (15.5) | 54 (14.6) | 0.063 |
| Moderate | | 172 (40.4) | 137 (37.1) | |
| Major | | 129 (30.3) | 142 (38.5) | |
| Extreme | | 59 (13.8) | 36 (9.8) | |
| In-hospital mortality n (%) | 426|369 | 58 (13.6) | 34 (9.2) | 0.053 |

$N_0$: Number of patients with available data during the pre-pandemic period; $N_1$ Number of patients during the pandemic period.

**Table 2. Time invested in each of the phases of prehospital and hospital care.**

| Pre-hospital times (minutes) | | | | | |
|---|---|---|---|---|---|
| | Pre-pandemic | | Pandemic | | |
| | N | Median (IQR) | N | Median (IQR) | p value |
| Step 1: 112 Call-Dispatch | 506 | 4.4 (2.8;7.2) | 436 | 4.8 (3.0;8.2) | 0.022 |
| Step 2: Dispatch-Mobilization | 506 | 1.9 (0.8;3.2) | 441 | 2.4 (1.2;4.1) | <0.001 |
| Step 3: Mobilization-Intervention | 492 | 9.6 (6.8;13.3) | 427 | 9.5 (6.9;14.0) | 0.750 |
| Step 4: Intervention-Departure to hospital | 478 | 33.5 (27.9;41.2) | 407 | 37.6 (29.9;46.1) | <0.001 |
| Step 5: Departure-Arrive at hospital | 439 | 15.2 (9.0;25.9) | 372 | 15.6 (9.5;23.8) | 0.923 |
| Step 1–5: Call-Arrival at the hospital | 449 | 72.3 (59.9;86.6) | 383 | 75.4 (64.0;91.4) | <0.001 |
| Hospital times (days) | | | | | |
| | Pre-pandemic | | Pandemic | | p value |
| Step 6 Length of hospital stay | 378 | 6.0 (2.9;11.6) | 323 | 6.2 (3.2;12.4) | 0.281 |

higher proportion of males (54.8% vs. 47.3%, difference 6.5%; 95% CI: 0.2; 12.78; p value = 0.04). No statistically significant differences were detected in the vital signs collected, in the Charlson comorbidity index, the degree of severity, the predicted risk for mortality or the proportion of positives on Madrid Direct scale. The proportion of patients who underwent a prehospital electrocardiogram was reduced by almost 10%. The in-hospital mortality changed from 14% to 9%, without statistical significance.

During the pandemic, all the periods of time increased to a greater or lesser extent (Table 2 and Fig 3). Among the prehospital intervention times, the call management time by the coordinating centre stands out (step 1), as it increased by 9% (4.8 vs. 4.4 minutes; difference 0.4; 95% CI: -0.11; 0.91; p value = 0.02), and the time on scene (step 4) increased by 12% (37.6 vs. 33.5 minutes, average difference 4.1; 95% CI: 2.49; 5.93; p value = <0.01). Median length of hospital stay was prolonged by 3% (6.2 days vs. 6.0; average difference 0.2; 95% CI:-1.28;0.91; p value = 0.28). These analyses were adjusted by age and did not make any significant difference (data not shown). In the interrupted time series, the direct change produced by the effect of the pandemic in the time "on scene" is observed, although with a non-significant slope (slope -0.01 95% CI: -0.04; 0.02), and it decreased with the passage of time (Fig 4).

Neither the concordant diagnosis of SUMMA 112 with hospital discharge nor the analysis of diagnostic discrepancies by MBDS pathology showed significant differences between the two periods (Table 3). No significant differences were found in the percentage of patients treated with IVT or MT. There were also no significant differences in in-hospital mortality, although a decrease was observed in the "pandemic" group (difference 4.4%; 95%IC: 0.0; 8.8; p-value = 0.05) (Table 1).

## Discussion

To our knowledge, this is the first study to analyse in detail the impact of COVID-19 in the metrics for stroke care at the pre-hospital setting of the SC at regional level. Other articles reporting increase in emergency calls, the decrease in stroke alerts, have generally studied

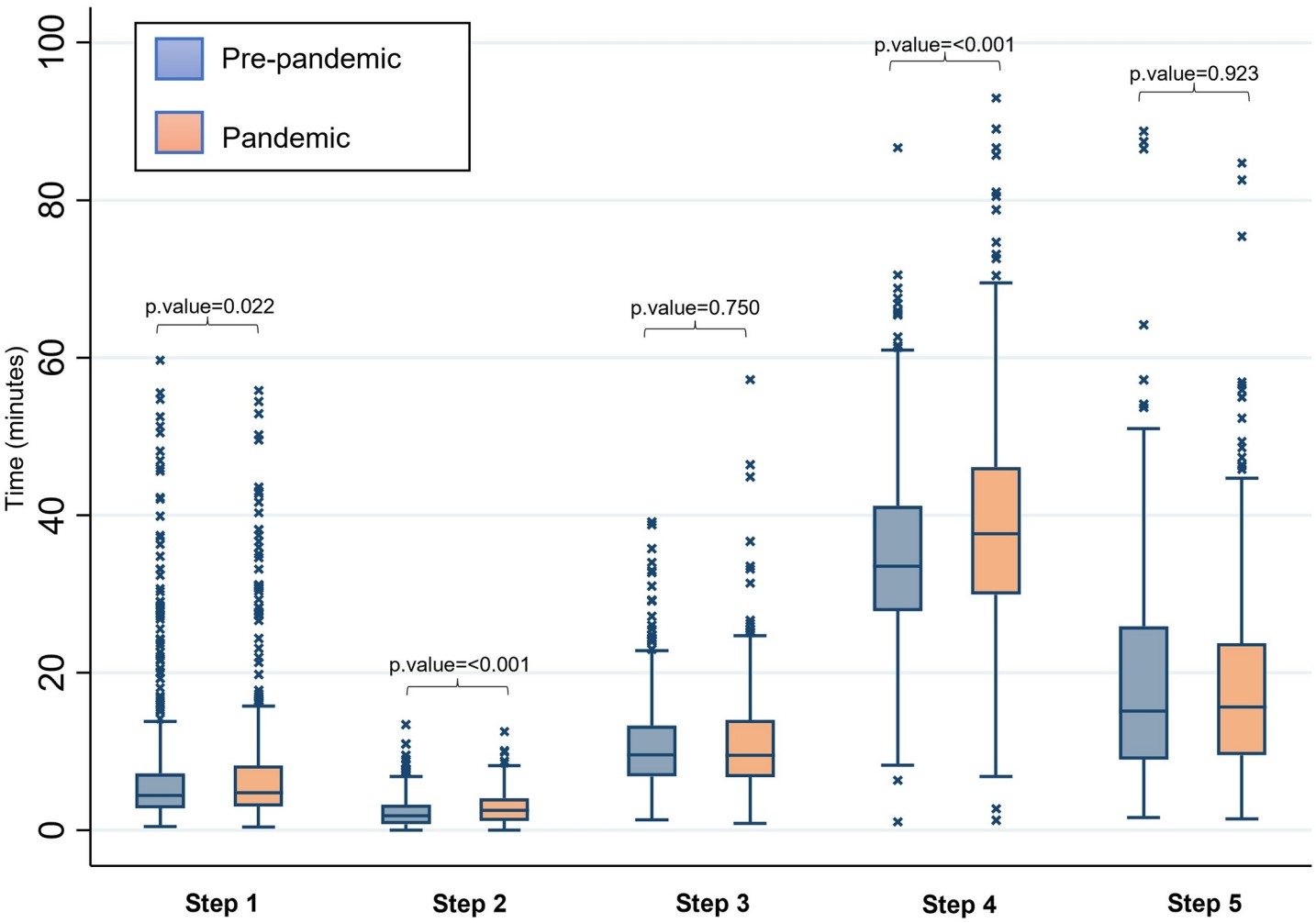

**Fig 3. Box chart with prehospital times in each of the health care phases.**

times, or have focused on the hospital setting [13, 29–35], but we have not found one that have linked pre-hospital data with a hospital administrative registry to capture the main aspects related to EMS SC performance in a comprehensive approach.

Our main finding was that while the transit times of the ambulances (steps 3 and 5) did not suffer significant variations (although the confinement measures positively influenced the traffic during the first wave), the response times of the 112 call centre (step 1) and on-scene times (step 4) increased more than 10% during the first wave of the pandemic (Fig 3). Median length of hospital stay also increased, albeit to a lesser extent (<4%).

Clinical practice guidelines on AS recommend call centre response times of less than 1 minute and less than 15 minutes on scene [36]. The correct operation of the call centre is essential for the urgent management of pathologies that can endanger life [37]. The overcrowd of calls from patients with symptoms of COVID-19 could saturate the SUMMA-112 call centre at certain times, causing a high level of interference with the attention given to other urgent calls. During the worst days of the first wave, the number of calls received in the same period during the previous year tripled. Consistently, they increased by 212% in Lausanne (Switzerland), increased by 225% in Paris, and tripled in Emilia Romagna (Italy) and Catalonia (Spain) [13,

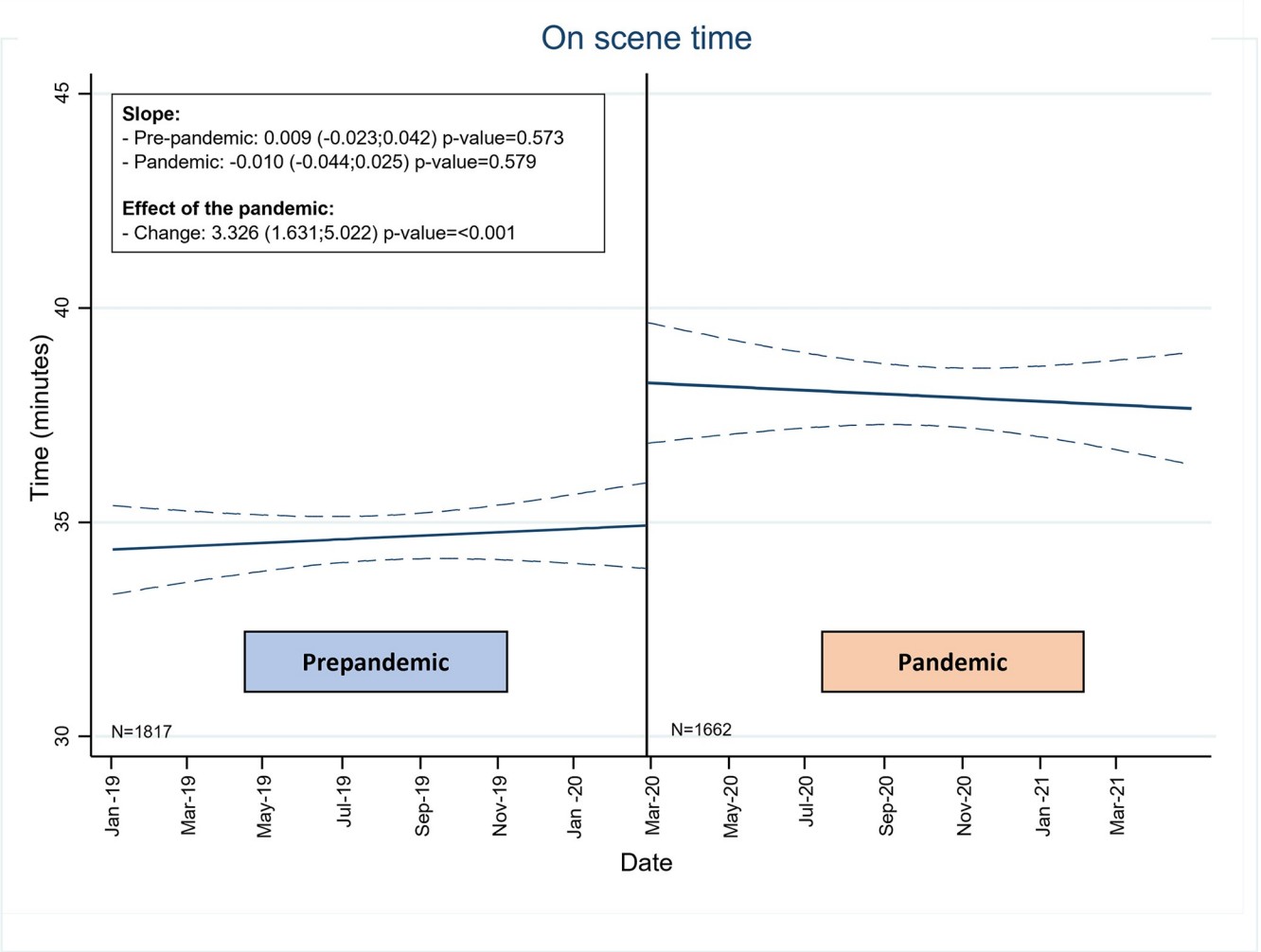

**Fig 4. Interrupted time series with the "on-scene" time of SUMMA 112.**

29, 38]. Until an alternative number was established by the health authorities for consultations related to COVID-19, many of the patients used 112 to resolve their doubts and request assistance. Therefore, in our opinion, it is very important to establish call overflow control methods to avoid excessively prolonged waiting to be attended to. Using a self-diagnosis application or transferring the avalanche of calls from patients with questions about COVID-19 to another number that is not the emergency number are potential solutions that have demonstrated their effectiveness [30].

With regards to the time on the scene (step 4), the most relevant change was the use of personal protective equipment (PPE) against biological risks that had to be placed just before attending to the patient. The placement process must be orderly, sequential, and supervised by another member of the team to avoid errors. In the first days, the lack of dexterity in the procedure could cause an increase in the time recorded. A study in Western Pennsylvania involving 22 emergency agencies (urban, suburban and rural) showed an increase of about 10% (15.7 ± 9.5 to 18.3 ± 10.8) for all diseases [31]. Even in minimally affected regions such as Okayama, with only 16 cases of COVID-19 during the first wave, there was an increase in scene time similar to ours, so we believe that it is an independent effect of the incidence of COVID-

**Table 3. Diagnoses at hospital discharge.**

| STROKE diagnosis | Pre-pandemic N = 426 | Pandemic N = 369 | p value |
|---|---|---|---|
| Stroke code | | | |
| Diagnostic Accuracy | 378 (88.7%) | 316 (85.6%) | 0.191 |
| (EMS–Hospital concordant proportion) | | | |
| Non-concordant proportion | 48 (11.2%) | 53 (14.4%) | |
| Of the non-concordant | | | |
| Brain and meningeal infections | 4 (8%) | 3 (6%) | 0.798 |
| Other infections: Sepsis, pneumonia, empyema, retroperitoneal abscess | 4 (8%) | 6 (11%) | |
| Other cancers and tumours | 2 (4%) | 3 (6%) | |
| Brain and meningeal and metastatic tumours | 5 (10%) | 7 (13%) | |
| Hyponatremia, diabetes | 2 (4%) | 1 (2%) | |
| Localized amyloidosis, delirium, dysautonomias, Horner, orthostatic hypotension, cysts | 11 (23%) | 11 (21%) | |
| Epilepsies and seizures | 10 (21%) | 17 (32%) | |
| Migraines and headaches | 2 (4%) | 1 (2%) | |
| AMI, PTE, valvopathies, arrhythmias, COPD, asthma, atelectasis, liver failure, autoimmune | 8 (17%) | 4 (8%) | |

19 [32]. In the analysis of the interrupted time series, a progressive reduction was observed (Fig 4). We believe that this later reduction reflects the learning curve effect. Training in these skills not only reduces the time it takes for the teams to put on and remove the PPE but also increases confidence that the process is being done with the required level of safety.

Although a few groups report large prolongations, mainly driven by the hospital saturation delaying the diagnostic and therapeutic processes [33, 39], most report decreases (which may be explained by the need to ensure available beds to attend more patients coming from the emergency departments) or no differences [40–43]. In our study we found a slight increase in length of stay (near 3 hours), although we did not conduct an analysis on the specific reasons for this mild increase.

SUMMA 112 maintained a percentage of greater than 85% of diagnostic accuracy that did not show significant variations during the pandemic. The increase in the number of transfers of patients with stroke mimics would have meant a significant overload to the system already decimated by the pandemic itself. EMS from other countries have found a greater proportion of stroke mimics [44–46]. The strength of our protocol lies in the dispatch of these missions to teams with highly qualified health personnel (doctors and nurses), who are able to distinguish the neurological symptoms that correspond to AS with greater precision, as occurs in other countries with similar EMSs [47, 48].

Regarding the treatment of ischaemic AS, despite the saturation of the health system during the first wave, the Stroke Network of the CM has been able to maintain the same proportion of IVT and MT. This finding is consistent with that published by our stroke network at hospital level [49, 50]. Many of the studies already published reflect a lower percentage of reperfusion treatments, which is more striking in centres with a high volume of admissions for COVID-19 [34]. A French study reported a decrease of more than 20% in MT, with an increase of more than 10% in door-to-groin times [35]. Our results show the great resilience of our network since, in circumstances of very high care pressure, it has been able to maintain similar proportions of patients treated.

Although not statistically significant, the reduction in in-hospital mortality is striking. We did not find a simple explanation for the reduction among patients with stroke during the first wave of the pandemic. Similar findings have been published in the literature for common acute conditions [41]. There are no data in the estimation of severity of both groups that

explain it (neither in the Charlson index nor in the other indices analysed). One possibility, consistent with the finding of a lower mean age during this period, is that many elderly patients did not seek assistance when they had stroke symptoms or that they were less likely to be not transferred to hospitals during the pandemic due to the overcrowd of the health services [51, 52]. Another hypothesis could be the competing effect between COVID-19 and stroke, i.e. older patients who were at risk for suffering stroke might be affected first by COVID-19 resulting in death.

The use of the data obtained by the SUMMA 112 linked with the hospital administrative data extracted from the discharge report of the patients using the MBDS has allowed us to obtain information on the aspects of urgent care of the AS in the CM that have been most widely seen to be affected during the first wave of the pandemic, as well as its effect on therapeutic management and prognosis.

Our study has some limitations. First, it refers only to strokes treated by SUMMA 112, not considering patients who are admitted to the hospital by other means. However, it is important to note that these are usually the most severe stroke patients and, therefore, those who would be most affected by a deviation in the SC protocol [53, 54]. Second, hospital data could not be collected in all patients. Most of the records lost in the MBDS were due to the absence of the HIN, mainly in those patients who come from other autonomous communities, from private insurers, or because such data could not be collected by the SUMMA 112. In these cases, only the prehospital variables were analysed. However, this affected a small proportion of our sample, and we believe that it does not detract from the results obtained, since our main objective was to analysis the time metrics at prehospital setting and all those patients were included in that analysis. Diagnostic accuracy is reduced to 74% and 70% (prepandemic and pandemic groups respectively) if we take into account all SC patients, although this includes patients without HIN, without a CMBD record or patients who may have died before the hospital admission. Third, the accuracy of the MBDS for diagnoses and treatments is a limitation. There are studies that have calculated its sensitivity above 82% for AS and a specificity above 95% [55]. We have not found literature analysing the validity with respect to treatments and therapeutic procedures. However, we believe that the use of the MBDS is an alternative when the records that professionals fill in manually are incomplete due to the overload of work that the pandemic has brought. In addition, we cannot rule out the possibility of an immortality bias, but if it were to arise, we believe it would not be significant given the similarity between pandemic and prepandemic populations.

## Conclusions

The analysis of SC in the CM during the first wave of the pandemic leads to the conclusion that there has been a negative impact on the time metrics of stroke code, with a 10% increase in the spent on the telephone, time spent on stage, time to admission and length of hospital stay. The diagnostic accuracy of EMS professionals was not significantly affected, showing one of the strengths of the SUMMA 112 EMS teams. Finally, the proportion of patients treated with IVT or MT was not reduced, demonstrating the great resilience of the Stroke Network of the CM.

## Supporting information

**S1 Table. List of MBDS diagnosis of AS.**
(DOCX)

**S1 File. Members of the Madrid stroke network.**
(DOCX)

**S2 File. Madrid direct scale.** Description of the M-Direct scale.
(DOCX)

**S3 File. Alternative language manuscript.** Manuscrito en Español.
(DOCX)

## Acknowledgments

To María José Medrano Martínez, Gonzalo Bayo Martínez, and Juan Antonio Pajuelo Ayuso at the IT Department of SUMMA 112 for their computer development of the DELFOS application, an invaluable help for data extraction. To the nursing coordination board at SUMMA 112. To the members of the documentation and diagnostic coding department of the La Paz and Doce de Octubre hospitals in Madrid for their help in deciphering the MBDS. To the members of the Madrid Stroke Network (listed below) for they for their support and work in the collection of hospital data. To the Consejería de Sanidad de la Comunidad de Madrid, for its strong support for research. To the Foundation for Biosanitary Research and Innovation in Primary Care (FIIBAP) for its financial support and assistance. To all the professionals of SUMMA 112 and the hospitals of the Stroke Network who, in these extraordinary circumstances, have maintained their commitment to their patients and to the health institution with heroic criteria.

Madrid Stroke Network multidisciplinary working group members as of 2021 are listed in S1 File. Lead author: Exuperio Díez Tejedor (Department of Neurology and Stroke Centre, Hospital Universitario La Paz), contact email: exuperio.diez@salud.madrid.org.

## Author Contributions

**Conceptualization:** Nicolás Riera-López, Carmen Cuadrado-Hernández, Emmanuel Pelayo Martínez-González, Alicia Villar-Arias, Fátima Gutiérrez-Sánchez, Pablo Busca-Ostolaza, Eduardo Montero-Ruiz, Exuperio Díez-Tejedor, Javier Zamora, Blanca Fuentes-Gimeno.

**Data curation:** Andrea Gaetano-Gil, José Martínez-Gómez, Nuria Rodríguez-Rodil, Borja M. Fernández-Félix, Carmen Cuadrado-Hernández, Emmanuel Pelayo Martínez-González, Eduardo Montero-Ruiz, Javier Zamora.

**Formal analysis:** Andrea Gaetano-Gil, Borja M. Fernández-Félix, Jorge Rodríguez-Pardo, Eduardo Montero-Ruiz, Exuperio Díez-Tejedor, Javier Zamora.

**Funding acquisition:** Nicolás Riera-López, Alicia Villar-Arias, Fátima Gutiérrez-Sánchez, Pablo Busca-Ostolaza.

**Investigation:** Nicolás Riera-López, Andrea Gaetano-Gil, José Martínez-Gómez, Borja M. Fernández-Félix, Jorge Rodríguez-Pardo, Pablo Busca-Ostolaza, Eduardo Montero-Ruiz, Exuperio Díez-Tejedor, Javier Zamora, Blanca Fuentes-Gimeno.

**Methodology:** Nicolás Riera-López, Andrea Gaetano-Gil, José Martínez-Gómez, Nuria Rodríguez-Rodil, Borja M. Fernández-Félix, Jorge Rodríguez-Pardo, Eduardo Montero-Ruiz, Exuperio Díez-Tejedor, Javier Zamora, Blanca Fuentes-Gimeno.

**Project administration:** Nicolás Riera-López.

**Resources:** Nicolás Riera-López, Carmen Cuadrado-Hernández, Emmanuel Pelayo Martínez-González, Alicia Villar-Arias, Fátima Gutiérrez-Sánchez, Pablo Busca-Ostolaza.

**Software:** Andrea Gaetano-Gil, José Martínez-Gómez, Nuria Rodríguez-Rodil, Borja M. Fernández-Félix, Javier Zamora.

**Supervision:** Jorge Rodríguez-Pardo, Eduardo Montero-Ruiz, Exuperio Díez-Tejedor, Javier Zamora, Blanca Fuentes-Gimeno.

**Validation:** Nicolás Riera-López, Andrea Gaetano-Gil, José Martínez-Gómez, Nuria Rodríguez-Rodil, Borja M. Fernández-Félix, Blanca Fuentes-Gimeno.

**Visualization:** Nicolás Riera-López, Andrea Gaetano-Gil, Borja M. Fernández-Félix, Javier Zamora.

**Writing – original draft:** Nicolás Riera-López, Jorge Rodríguez-Pardo, Eduardo Montero-Ruiz, Exuperio Díez-Tejedor, Javier Zamora, Blanca Fuentes-Gimeno.

**Writing – review & editing:** Nicolás Riera-López, Andrea Gaetano-Gil, José Martínez-Gómez, Nuria Rodríguez-Rodil, Borja M. Fernández-Félix, Jorge Rodríguez-Pardo, Carmen Cuadrado-Hernández, Emmanuel Pelayo Martínez-González, Alicia Villar-Arias, Fátima Gutiérrez-Sánchez, Pablo Busca-Ostolaza, Eduardo Montero-Ruiz, Exuperio Díez-Tejedor, Javier Zamora, Blanca Fuentes-Gimeno.

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
