## [Decision Letter · Decision Letter 0]

1 Aug 2022

PONE-D-22-10142The COVID-19 pandemic effect on the prehospital Madrid Stroke code metrics and diagnostic accuracyPLOS ONE

Dear Dr. Riera Lopez,

Thank you for submitting your manuscript to PLOS ONE. After careful consideration, we feel that it has merit but does not fully meet PLOS ONE’s publication criteria as it currently stands. Therefore, we invite you to submit a revised version of the manuscript that addresses the points raised during the review process.

Thanks for conducting this study,  Please consider all given important comments by reviewers  in revised version,  particularity  possible bias

We look forward to receiving your revised manuscript.

Kind regards,

Hamid Reza Baradaran, M.D., Ph.D.,

Academic Editor

PLOS ONE

Journal Requirements:

I have read the journal's policy and the authors of this manuscript have the following competing interests: Nicolás Riera-López and Jorge Rodríguez-Pardo de Donlebún have received payments for training courses from the Angels Initiative (Boheringer Ingelheim). The other authors do not report any conflicts of interest.

5. One of the noted authors is a group or consortium Madrid Stroke Network. In addition to naming the author group, please list the individual authors and affiliations within this group in the acknowledgments section of your manuscript. Please also indicate clearly a lead author for this group along with a contact email address.’ 

Additional Editor Comments:

Thanks for conducting this study, Please consider all given important comments by reviewers in revised version, particularity possible bias

Reviewers' comments:

Reviewer's Responses to Questions

**Comments to the Author**

1. Is the manuscript technically sound, and do the data support the conclusions?

Reviewer #1: Yes

Reviewer #2: Yes

2. Has the statistical analysis been performed appropriately and rigorously? 

Reviewer #1: No

Reviewer #2: Yes

3. Have the authors made all data underlying the findings in their manuscript fully available?

Reviewer #1: Yes

Reviewer #2: Yes

4. Is the manuscript presented in an intelligible fashion and written in standard English?

Reviewer #1: Yes

Reviewer #2: No

5. Review Comments to the Author

Reviewer #1: Title: OK

Abstract: according to the definition of the methodology used in the paper, it seems this is not a quasi-experimental study; because the researcher did not have a role for exposure allocation (COVID-19). It is important to show the duration of each phase in the context of this section (material and method part of the abstract).

Introduction: OK

Material and method: this study is not a quasi-experimental one as explained above. I think it is necessary to define the study setting in more detail. This especially true for the very critical patients (who might have died before arriving at the hospital). As a matter of the fact, I am very concern about the matter of immortal time bias which may easily happen in such studies. Please define the method of estimating each time variable in more detail so that the method of the measurements can be reproduced in future studies. This is true for the diagnostic accuracy and severity status as well. In case of severity index, I think estimation of reliability is necessary in your own study. Is it possible to assess the number of repeated cases in each period of the study? This is because that in case of large number of these cases, autocorrelation consideration may be necessary for the analysis.

Results: There are a number of variables which are not addressed in the previous section (e.g., Electrocardiogram, temperature and …). According to the results shown in table 1 (lower mean age in pandemic era), I think it may be necessary to adjust the addressed results according to the age of the patients.

Reviewer #2: - Were the two groups similar in terms of age, sex, type and severity of stroke?

- Is the pattern of stroke occurrence in two groups the same?

- The manuscript need to basic scientific editing.

- Is it enough to study only the first wave of COVID-19 to reach such results? Isn't it better to evaluate the actions in the next waves?

- I think the type of study should be changed to observation before- after study.

- The code of ethics should be mentioned.

- Who answers the calls? Was the experience and skill of the respondents the same? It must be mentioned in the methodology.

- Hadn't the instructions changed in these two periods?

- The inclusion criteria for patients in unclear?

- It is better to write abnormal instead of non-normal.

- Why is the bootstrap method used to calculate the differences? and why has it been resampled 800 times? Which bootstrap method is used? Parametric or non-parametric? Be clearly mentioned.

- The P<0.01 is not valid. It should be written up to three decimal places or as p<0.001.

6. PLOS authors have the option to publish the peer review history of their article (what does this mean?). If published, this will include your full peer review and any attached files.

Reviewer #1: **Yes: **Babak Eshrati

Reviewer #2: No

---

## [Author Response · Author response to Decision Letter 0]

14 Sep 2022

Answers to editor and reviewer comments are stated in the attached document: "Response to reviewers v3"

---

## [Editor Report · Decision Letter 1]

26 Sep 2022

The COVID-19 pandemic effect on the prehospital Madrid Stroke code metrics and diagnostic accuracy

PONE-D-22-10142R1

Dear Dr. Riera Lopez,

We’re pleased to inform you that your manuscript has been judged scientifically suitable for publication and will be formally accepted for publication once it meets all outstanding technical requirements.

Kind regards,

Hamid Reza Baradaran, M.D., Ph.D.,

Academic Editor

PLOS ONE
---

## [Editor Report · Acceptance letter]

28 Sep 2022

PONE-D-22-10142R1 

The COVID-19 pandemic effect on the prehospital Madrid Stroke code metrics and diagnostic accuracy 

Dear Dr. Riera-López:

I'm pleased to inform you that your manuscript has been deemed suitable for publication in PLOS ONE. Congratulations! Your manuscript is now with our production department. 

Kind regards, 

on behalf of

Professor Hamid Reza Baradaran 

Academic Editor

PLOS ONE